# Dielectric multi-momentum meta-transformer in the visible

Lei Jin[1,9], Yao-Wei Huang [1,2,9], Zhongwei Jin[1,9], Robert C. Devlin[2], Zhaogang Dong [3], Shengtao Mei[1], Menghua Jiang[1], Wei Ting Chen [2], Zhun Wei[1], Hong Liu[3], Jinghua Teng [3], Aaron Danner[1], Xiangping Li [4], Shumin Xiao[5], Shuang Zhang [6], Changyuan Yu[1,7], Joel K.W. Yang [3,8], Federico Capasso[2]* & Cheng-Wei Qiu [1]*

Metasurfaces as artificially nanostructured interfaces hold significant potential for multi-functionality, which may play a pivotal role in the next-generation compact nano-devices. The majority of multi-tasked metasurfaces encode or encrypt multi-information either into the carefully tailored metasurfaces or in pre-set complex incident beam arrays. Here, we propose and demonstrate a multi-momentum transformation metasurface (i.e., meta-transformer), by fully synergizing intrinsic properties of light, e.g., orbital angular momentum (OAM) and linear momentum (LM), with a fixed phase profile imparted by a metasurface. The OAM meta-transformer reconstructs different topologically charged beams into on-axis distinct patterns in the same plane. The LM meta-transformer converts red, green and blue illuminations to the on-axis images of "R", "G" and "B" as well as vivid color holograms, respectively. Thanks to the infinite states of light-metasurface phase combinations, such ultra-compact meta-transformer has potential in information storage, nanophotonics, optical integration and optical encryption.

[1] Department of Electrical and Computer Engineering, National University of Singapore, 4 Engineering Drive 3, Singapore 117583, Singapore. [2] Harvard John A. Paulson School of Engineering and Applied Sciences, Harvard University, Cambridge, MA 02138, USA. [3] Institute of Materials Research and Engineering, A*STAR (Agency for Science, Technology and Research), 2 Fusionopolis Way, #08-03 Innovis, Singapore 138634, Singapore. [4] Guangdong Provincial Key Laboratory of Optical Fiber Sensing and Communications, Institute of Photonics Technology, Jinan University, Guangzhou 510632, People's Republic of China. [5] Ministry of Industry and Information Technology Key Lab of Micro-Nano Optoelectronic Information System, Harbin Institute of Technology, Shenzhen, Guangdong 518055, People's Republic of China. [6] School of Physics and Astronomy, University of Birmingham, Birmingham B15 2TT, UK. [7] Department of Electronic and Information Engineering, The Hong Kong Polytechnic University, Hung Hom, Kowloon, Hong Kong. [8] Singapore University of Technology and Design, 8 Somapah Road, Singapore 487372, Singapore. [9] These authors contributed equally to this work: Lei Jin, Yao-Wei Huang, Zhongwei Jin. *email: capasso@seas.harvard.edu; chengwei.qiu@nus.edu.sg

Metasurfaces composed of tailored nanostructures arranged two-dimensionally hold great capabilities to locally control light's phase, amplitude and polarization states at the subwavelength scale[1–7]. Due to this control capability, metasurfaces can provide specific transmission (T) and reflection (R) functions to work as planar photonic components carrying customized information. Such information can be unlocked via a distinguished field distribution reconstructed at the observed region. When an incident light $U_{inc}(x_0, y_0)$ impinges upon a metasurface $U_{meta}^{T/R}(x_0, y_0)$, the field distribution can be expressed by the convolution of $U_{meta}^{T/R}(x_0, y_0) U_{inc}(x_0, y_0)$ and an impulse response $h(x, y, z)$ that relates the fields at the metasurface. Therefore, the field distribution on the observation plane can be expressed as (more detail are shown in Supplementary Note 1):

$$U(x, y, z) = \iint\limits_{-\infty}^{+\infty} U_{meta}^{T/R}(x_0, y_0) U_{inc}(x_0, y_0) h(x - x_0, y - y_0, z) dx_0 dy_0$$

$$(1)$$

The diversity of metasurfaces' function $U_{meta}^{T/R}(x_0, y_0)$ serves as the base for realizing lenses[8–10], color[11–14], polarization filters[15,16] and holograms[17–20].

Multi-tasked metasurface is preferably desired for more compact nanophotonic devices, which could reconstruct multiple distinguished field distributions $U^{(n)}$ at observed regions. Based on Eq. 1, previously reported multi-tasked metasurfaces can be divided into three categories (shown in Fig. 1a–c). The first approach takes advantage of spatial separation as shown in Fig. 1a. By separating the observed regions $(S^{(n)})$[21,22] or interleaving subarrays $\left(S_0^{(n)}\right)$[23–29] specifically designed for each functionality on the metasurface, the spatial multiplexed metasurface can reconstruct different field distributions ($U^{(n)}$), but its efficiency is limited. The second approach resorts to adjusting functions $U_{meta}^{(n)}$ of the metasurface by changing polarizations[30–36], incident angles[37] or wavelengths[38] of beams as shown in Fig. 1b. By tailoring meta-atoms, the structurally multiplexed metasurface carries polarization-, angle- or wavelength-dependent responses $U_{meta}^{(n)}$ to reconstruct different field distributions ($U^{(n)}$). This method can achieve higher efficiency than the previous one, but the efficiency is still limited by the coverage of the required phases that meta-atoms need to achieve. Another approach, as shown in Fig. 1c, is based on OAM multiplexing chip to read out the pre-set incident light ($A_{inc}$)[39], which pre-generate various information-carrying multifocal beams arrays with corresponding OAM states. Each ring groove of the chip can out-couple a determined OAM order[40,41], which equivalently provides a "key" array to read out the distributions $U^{(n)}$ locked in the preset beams by the spatial light modulator (SLM)[39]. However, the outcoupling efficiency of slits is much smaller than that in the previous two methods and it needs the beam array to illuminate each meta-atom. So far, a majority of reported multi-tasked metasurface are resorting to the spatial freedom, structure complexity or pre-modulation with decoupler, and there is a lack of a convenient way to realize multi-tasked functionality by intrinsic properties of light.

In this paper, we report a transmission-type multi-momentum meta-transformer, which transforms the intrinsic phases of OAM ($l\hbar$) and LM ($k_0\hbar$) of the incidence light, into various distinct patterns in the same plane (Fig. 1d). The meta-transformer is made of the minimalist $TiO_2$ nano-fin array (Fig. 1g, h) and provides a phase profile $\psi_{meta}(x_0, y_0) = 2\varphi(x_0, y_0)$ for the right circularly polarized beam with spin angular momentum (SAM). The phase is based on geometric phase and $\varphi(x_0, y_0)$ is the orientation angle of nano-fins as a function of position. For a given fabricated metasurface, incident beams $U_{inc}(x_0, y_0)$ carrying OAM or LM are able to impart extra phase profiles $\psi_{OAM}(x_0, y_0)$ or affect impulse response $h(k_0)$ to the transmitted light to realize distinct patterns in the same plane right on axis. A multi-OAM phase retrieval algorithm is developed to design the OAM meta-transformer, which can "read out" the order of the incident vortex beam. To be more specific, an image which tells the order of the incident vortex beam will be reconstructed at a given plane under illumination of vortex beam with certain orders as shown in Fig. 1e. In addition a multi-LM phase retrieval algorithm is proposed to design a LM meta-transformer, that can reconstruct patterns "R", "G" and "B" in the same plane by illuminating it with red $\left(k_0^{(R)}\right)$, green $\left(k_0^{(G)}\right)$, and blue $\left(k_0^{(B)}\right)$ beams, respectively (Fig. 1f). This is due to the impulse response in Eq. 1 which is $k_0$-dependent. Moreover, such LM meta-transformer has demonstrated the capability to display vivid colorful images, thanks to the high-efficiency of the dielectric metasurface and the on-axis imaging feature for all wavelengths.

## Results

**Principle of multi-momentum meta-transformer.** The metasurface consists of a set of amorphous $TiO_2$ nano-fins arranged in square pixels on a quartz substrate (Fig. 2a). The pixel size is $325 \times 325$ nm$^2$ and the $TiO_2$ nano-fin is 80-nm-wide, 250-nm-long and 600-nm-high, which rotates in plane with an orientation angle $\varphi$. When a circularly polarized beam is normally incident on the metasurface from the side of quartz, the transmitted light converts to the opposite circular polarization (CP) and acquires a phase delay of $\pm 2\varphi$ (Supplementary Note 2). Lumerical FDTD solution is employed to optimize the geometrical parameters of nano-fins, such that the conversion from one CP to the opposite CP is efficient for the operation in broadband visible range (as shown in Fig. 2b).

Figure 2c presents the schematic of OAM meta-transformer. A collimated vortex beam with native phase profile $\psi_{OAM}(x_0, y_0)$ illuminates the metasurface with its own as-fabricated phase $\psi_{meta}(x_0, y_0)$ from the quartz substrate, and the transmitted beam carries the total phase profile $\psi_T(x_0, y_0) = \psi_{OAM}(x_0, y_0) + \psi_{meta}(x_0, y_0)$ on illuminating area (Supplementary Note 3). The metasurface's phase function $\psi_{meta}(x_0, y_0)$, which is fixed after the fabrication, is defined by the in-plane orientation of nano-fin array. The extra phase profile $\psi_{OAM}(x_0, y_0)$ from the incident vortex beam is dynamically changeable. As a degree of freedom of light, by feeding different spiral phases to the single-phase metasurface, one meta-transformer can, therefore, provide multiple different phase profiles, resulting in different patterns on the same plane.

The phase function $\psi_{meta}(x_0, y_0)$ of OAM meta-transformer is designed by the multi-OAM phase retrieval algorithm (Supplementary Note 4). The orientation angle of each nano-fin is determined by the retrieval phase $\psi_{meta}(x_0, y_0)$. The schematic diagram of the experimental setup is shown in Supplementary Fig. 3a and its specification can be found in Supplementary Note 6. As shown in Fig. 2d, at the given observation plane ($z = 60$ μm), a judiciously designed metasurface reconstructs an "apple" pattern under illumination with a collimated Laguerre Gaussian beam with $l^{(1)} = -5$, while it shows a "spider" pattern instead when the OAM of the incident beam is changed to $l^{(2)} = 5$. The reconstructed images restore the features of the designed pattern and the crosstalk is suppressed.

Figure 2e presents the physical principle of LM meta-transformer. The phase function $\psi_{meta}(x_0, y_0)$ of a metasurface is defined by the in-plane orientation of nano-fin array, which means that the designed metasurface is able to provide the dispersionless phase profile. In the Fresnel region, the impulse

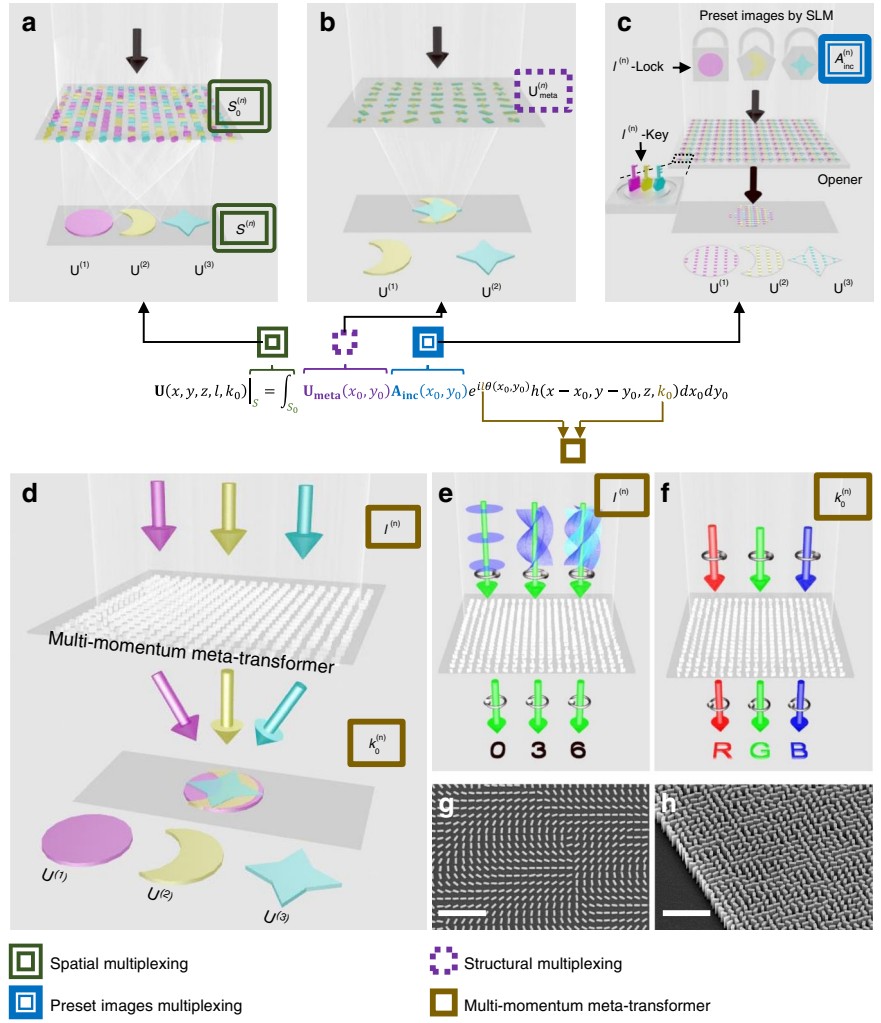

$$\mathbf{U}(x, y, z, l, k_0)\Big|_S = \int_{S_0} \mathbf{U_{meta}}(x_0, y_0)\mathbf{A_{inc}}(x_0, y_0)e^{i(\theta(x_0, y_0)}h(x - x_0, y - y_0, z, k_0)dx_0 dy_0$$

□ Spatial multiplexing ⠶ Structural multiplexing

□ Preset images multiplexing □ Multi-momentum meta-transformer

**Fig. 1** Comparative elaborations of multi-tasked metasurfaces (**a–c**) and our multi-momentum meta-transformer (**d–h**). **a** Schematic illustration of a spatially multiplexed metasurface[21–29]. The spatial multiplexed metasurface reconstructed several distinguished field distribution $U^{(n)}$ based on the spatial separations of metasurface $\left(S_0^{(n)}\right)$ or observed region ($S^{(n)}$). The $x_0$ and $y_0$ ($x$, $y$, and $z$) are the coordinated variables of the metasurface plane $S_0$ (observation region $S$). **b** Illustration of a structurally multiplexed metasurface[30–34,37,38]. The structurally multiplexed metasurface reconstructed several distinguished field distribution $U^{(n)}$ based on the metasurface function $U_{meta}^{(n)}$. **c** Schematic illustration of a multi-beam meta-opener[39]. The multi-beam meta-opener is formed by several kinds of "key arrays", which is used to read out the distributions $U^{(n)}$ carried by the preset incident beams. **d** Illustration of transmission-type multi-momentum meta-transformer. The multi-momentum meta-transformer decoder phase profile is implemented with TiO₂ nano-fin array with in-plane orientations on a quartz substrate. It controls multi-beams with different momenta ($l^{(n)}$ and $k_0^{(n)}$) to reconstruct corresponding field distributions $U^{(n)}$ **e** Schematic of the OAM meta-transformer. Under the illumination of vortex beams with right circular polarization (RCP), the decoder can generate distinct images at the same plane with left circular polarization (LCP). **f** Schematic of the LM meta-transformer. Under the illumination of different LM beams with RCP, the meta-transformer can generate distinct LM-dependent field distributions at the same region on optical axis. **g** Top-view scanning electron microscopy (SEM) images of a partial region of the fabricated TiO₂ nano-fins arrays. Scale bar: 2 μm. **h** Oblique-view SEM image. Scale bar: 2 μm. Each TiO₂ nano-fin represents a phase pixel as defined in the meta-transformer

response $h$ of Eq. 1 is

$$h(x, y, z) = \frac{e^{ik_0 z}}{i\lambda z} e^{i\frac{k_0}{2z}[x^2 + y^2]} \qquad (2)$$

Therefore, when collimated monochromatic beams illuminate the metasurface, this dispersionless phase profile $\psi_{meta}(x_0, y_0)$ can control the monochromatic beams with different LMs to achieve different E-field distribution in a given plane (shown in Fig. 2e).

To realize the LM meta-transformer, the key idea is to reconstruct LM-dependent patterns in the same plane. In the visible region, the monochromatic beams with different LMs represent different colors. So three-primary colors (red 633 nm, green 532 nm, and blue 488 nm) are chosen in this work. Based on the $k_0$-dependent impulse response, the multi-LM phase

retrieval algorithm (Supplementary Note 5) is deployed here to calculate the phase distribution $\psi_{meta}(x_0, y_0)$. Hence, the wavelength-dependent patterns are reconstructed in the same plane (at $Z_0$) and the unwanted patterns are moved out of the observed plane. In this design, based on the retrieval phase profile $\psi_{meta}(x_0, y_0)$, the nano-fins ($600 \times 600$) with rotation angles are fabricated on a total area of $192 \times 192$ μm². The schematic diagram of the experiment setup of color meta-hologram is shown in Supplementary Fig. 3b and the specification of the experiment setup can be found in Supplementary Note 6. The image is captured by a CCD camera. Figure 2f shows the holographic images of letter patterns "R","G","B" at a certain distance ($z = 195$ μm), corresponding to the RGB color incident beams (633 nm, 532 nm, and 488 nm). The images corresponding

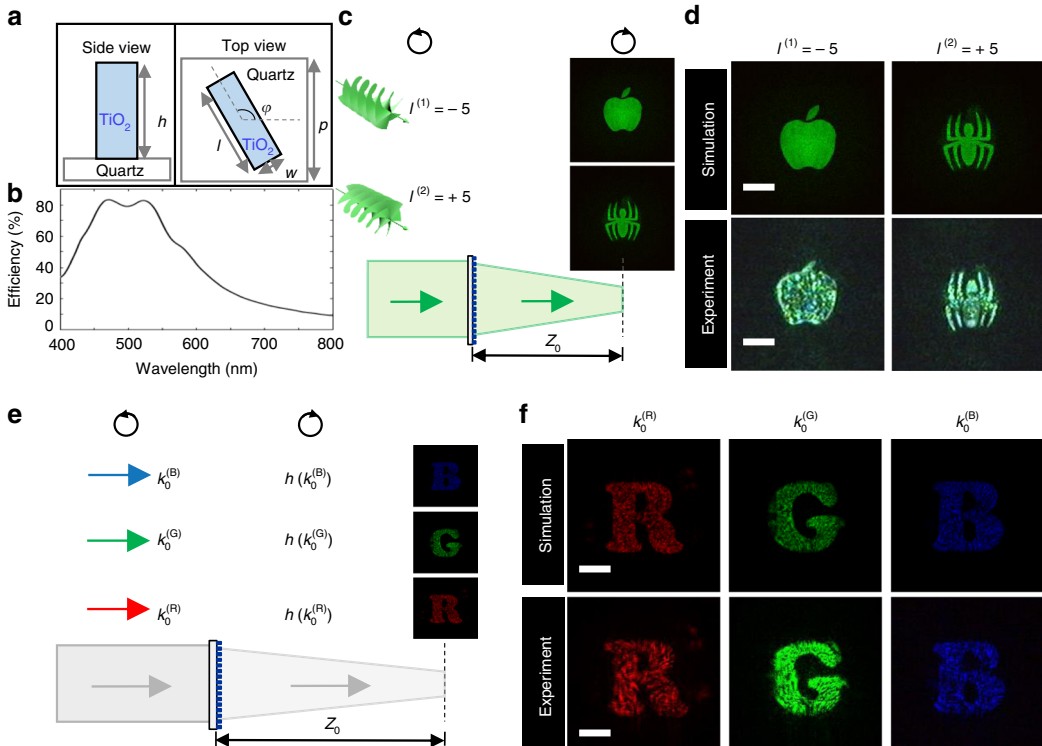

**Fig. 2** Principle and demonstration of multi-momentum meta-transformer. **a** Geometry of the designed unit cell structure representing one pixel in the meta-transformer, with the periodicity of 325 nm. The $TiO_2$ nano-fin parameters are $w = 80$ nm, $l = 250$ nm, and $h = 600$ nm. The in-plane rotating angle $\varphi$ of nano-fin will introduce the geometric phase of $2\varphi$ for the incident beam with RCP. **b** Measured conversion efficiency of the meta-transformer. The conversion efficiency is defined as the optical power of the transmitted light with opposite CP divided by the incident optical power. **c** Design principle of OAM meta-transformer. Under the illumination of vortex beam 1 ($l^{(1)} = -5$) with RCP, the transmitted beam with opposite CP carries the total phase profile $\psi_{OAM=-5}(x_0, y_0) + \psi_{meta}(x_0, y_0)$ and reconstructs "apple" pattern in the observation plane. When using vortex beam 2 ($l^{(2)} = 5$) with RCP, the total phase profile of the transmitted beam is $\psi_{OAM=5}(x_0, y_0) + \psi_{meta}(x_0, y_0)$, which causes the reconstructed pattern change to a spider-shaped pattern. **d** Simulated (top) and measured (bottom) reconstructed patterns by vortex beam 1 ($l^{(1)} = -5$) (left) and vortex beam 2 ($l^{(2)} = 5$) (right). Scale bar: 20 μm. **e** Design principle of LM meta-transformer. Under the illumination of right circularly polarized beam with $LM = k_0^{(R)}\hbar$, the transmitted beam with opposite CP carry the dispersionless phase profile of metasurface $\psi_{meta}(x_0, y_0)$. Due to the convolution of $\mathbf{U_{inc}}(x_0, y_0)exp(i\psi_{meta}(x_0, y_0))$ and impulse response $h(x, y, z, k_0^{(R)})$, the transmitted beam reconstructs patterns at the observation plane. Because the impulse response $h$ is $k_0$-dependent, by changing LM of incidence, the reconstructed images "R", "G" and "B" components can be shifted to one identical plane ($z = Z_0$). **f** Simulated (top) and experimental (bottom) reconstruction of three-primary color holograms at the imaging plane. Scale bar: 20 μm. The original "spider" image was obtained from PNG image website

to the RGB color incident beams are independent from each other, and the cross-talk among different LMs is eliminated, due to $k_0$-dependent impulse response.

## Discussion

The OAM meta-transformer can now work as an OAM displayer, as demonstrated in Fig. 3a. This meta-transformer is formed by $300 \times 300$ nano-fins on the total area $97.5 \times 97.5$ μm². As shown in Fig. 3b, the designed OAM meta-transformer reconstructs the patterns "0", "3" and "6" with the incident $0^{th}$-, $3^{rd}$- and $6^{th}$-order vortex beams, respectively. Compared with Fig. 2d, the OAM meta-transformer in Fig. 3b has lower phase difference among the incident vortex beams and encodes more OAM states, which may reduce the performance. Nevertheless, the reconstructions in Fig. 3b are clearly recognizable. The reconstructed patterns present the incident beam's topological charge values, which suggests a potential application of OAM topological charge displayer (OAM detection by direct reading) with such OAM meta-transformer.

Based on $k_0$-dependent impulse response, the LM meta-transformer generates LM-dependent patterns on the identical position (Fig. 2e, f). Under illumination of red, green and blue beams simultaneously, these generated patterns overlapping each

other enable the realization of colorful holographic images (Supplementary Note 7). The colorful holographic images contain not only three-primary colors (RGB) but also their secondary colors (cyan, magenta, yellow, as well as white). Moreover, the LM meta-transformer can also support complex patterns with gradient color (in Fig. 4a, b). Figure 4b reports the simulation and experimental results for each color and their superposition. These results indicate the accurate spatial control of the reconstructed images, and the crosstalk among different LMs is eliminated. The key feature for such gradient color image is that the color gamut spans the whole color triangle. Therefore, the LM meta-transform has the capability to display vivid colorful image.

The capability is a critical issue for the multi-tasked metasurface. Based on the multi-momentum phase retrieval algorithm provided in this work, a single meta-transformer is able to support 5 OAM states ranging from −8 to 8 or 6 LM states in the visible region (Supplementary Note 8). With the help of the polarization-dependent response[21,42,43] of $TiO_2$ nano-fins, the number of patterns encoded in a meta-transformer can increase to 9 for OAM or 11 for LM. Furthermore, the capability of a meta-transformer can also be improved by increasing the number of units[25], and widening the multi-momentum state region. Besides, by considering the donut shapes of vortex beams, introducing random phase mask[44], and combining the LM and

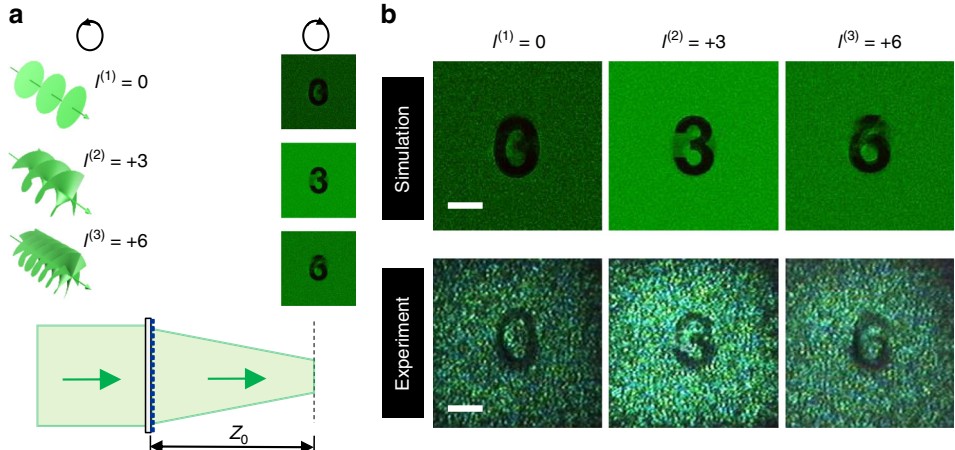

**Fig. 3** OAM meta-transformer for the read-out of three OAM states. **a** Schematic of OAM meta-transformer designed on three OAM states. This meta-transformer is designed to reconstructed "0", "3" and "6" patterns at $l^{(n)} = 0$, 3, and 6, respectively. **b** Simulated (top) and measured (bottom) reconstructed patterns. Scale bar: 20 μm

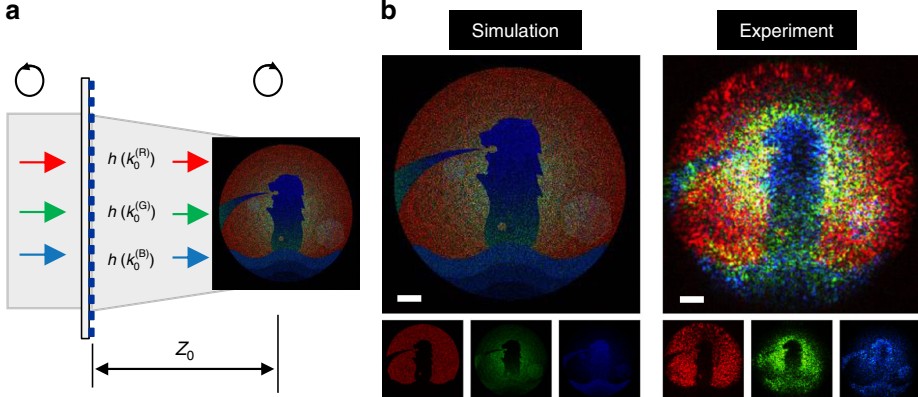

**Fig. 4** Reconstruction of color images. **a** Schematic illustration of reconstruction of gradient color image. **b** Intensity profiles corresponding to red, green, and blue beams and the graded color holographic image. Scale bar: 30 μm. The original "merlion" image was obtained from Vecteezy.com

OAM simultaneously in the phase retrieval algorithm, the multi-momentum meta-transformer has the potential to remarkably increase the carried information states.

In summary, we report a synergetic strategy of engaging OAM and LM of the incident beam with a single-phase metasurface, which significantly enriches the output of reconstructed patterns. OAM introduces the additional phase information from the incident beam, while the LM affects the phase information of impulse response. These two momentum degrees of freedom, in principle lead to infinite combined states, which enables metasurface with a large number of functions and may lead to new opportunities in 3D imaging, anti-counterfeiting, optical communication, and real-time detection.

## Methods

**Numerical simulation**. The amorphous $TiO_2$ nano-fins were optimized by Lumerical FDTD Solution (a commercial software). In this simulation, $TiO_2$ nano-fins with measured refractive index were placed on quartz substrate. Two sources polarized along $x$- and $y$-axes with a 90° phase shift was used to form the right circularly polarized incident beam, and this beam illuminated $TiO_2$ nano-fins from the substrate side. The periodic boundary condition was used along $x$- and $y$-directions, and the perfect matched layer (PML) was chosen along $z$-direction.

## Data availability

The data that support the findings of this study are available from the corresponding author upon reasonable request.

## Code availability

The codes that support the plots within this paper and other findings of this study are available from the corresponding authors upon reasonable request.

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

## Acknowledgements

This work was supported from the National Research Foundation, Prime Minister's Office, Singapore under its Competitive Research Program (CRP award NRF CRP15-2015-03). X.L. acknowledges the support from National Natural Science Foundation of China (NSFC) (Grant 61522504), Guangdong Provincial Innovation and Entrepreneurship Project (Grant 2016ZT06D081). F.C., Y.W.H., and W.T.C. acknowledge the financial support in part by the Air Force Office of Scientific Research (Grant Nos. MURI FA9550-14-1-0389 and FA9550-16-1-0156) and King Abdullah University of Science and Technology (KAUST) Office of Sponsored Research (OSR) (Award No. OSR-2016-CRG5-2995).

## Authors' contribution

L.J. and C.W.Q. conceived the idea. L.J., Z.J., S.M., M.J., and C.W.Q. designed the nanostructures and did the optical characterization of the hologram. Y.W.H. and R.C.D did the nanofabrication of nanostructures. Z.W. provided the expertise knowledge on the phase retrieval algorithm. Z.D., W.T.C., H.L., J.T., A.D., X.L., S.X., S.Z., C.Y., and J. K.W.Y. participated in the discussions and contributed with valuable suggestions for the multi-momentum meta-transformer. The paper was written by L.J. with inputs from Y.W.H, Z.J., S.M., F.C., and C.W.Q. C.W.Q. supervised the project. All authors analyzed the data, read and corrected the paper before paper submission.

## Competing interests

The authors declare no competing interests.
