## [Peer Review File · Nature Communications]

Reviewers' Comments:

Reviewer #1:

Remarks to the Author:

Optical metasurface devices, ultrathin inhomogeneous media with planar nanostructures that can manipulate light propagation in a desirable manner, leading to the development of many optical devices with unusual functionalities. Jin and colleagues propose and experimentally demonstrated a multi-momentum metasurface transformer based on the combination of intrinsic multi-phase of light, orbital angular momentum (OAM) and linear momentum. Although much efforts have been put in the generation, detection and manipulation of OAM beams, this work is interesting and can stimulate new thinking. The results presented in the manuscript seem to be correct, the experimental evidence provided support the conclusions, the manuscript is mostly clear. Regarding the significance of the results, I believe this manuscript will be of interest to specialists within the relevant fields. Thus, it seems to me that the manuscript can be accepted in this journal if the comments below can be fully addressed.

1. In Fig.2b, the calculated efficiency of the metasurface is very high. However, the signal-to-noise ratio in the Fig. 3b is very bad. I am not sure about the experimental value.
2. The performance of the devices. In Fig. 2d, the experimental results are better than those in Fig.3b. Is this related to the selection of the OAM? For example, the two topological charges with same absolute values but opposite signs ($l=-5$ and $+5$) are better than two positive values (1 and 3) or negative values.
3. The crosstalk issue. In Fig.3b, I can see other patterns in the OAM selective patterns. How many patterns can be encoded in the metasurface? I am sure there is a limit.
4. The generation of OAM beams is very important for device characterization. More detailed specifications of the SLM should be provided.
5. The reconstructed color images in Figure 4 and Supplementary Figure 4 are not attractive. I would like to encourage the authors to comment on their results. Our readers deserve to know them.
6. The overall language in the manuscript should be polished. There are also some typos, which need to be corrected.

Reviewer #2:

Remarks to the Author:

All-dielectric metasurfaces are in a big trend now towards ultimate control over the polarization state, focus points, orbital angular momentum, etc. of light beams. What is important that the thickness of the metasurfaces is comparative or even less than the wavelength, providing ultracompact tools for light manipulation. The paper well advances the subject of light control reporting on multiplication of actions: now orbital angular momentum can be combined with light's linear momentum. Certainly, such idea confirmed in direct experiments warrants publication in Nature Communications. I found few not clear wording and explanations/definitions that should be improved for the sake of the readers. Otherwise I don't need to review it again after such minor corrections.

1. "The majority of multi-tasked metasurface encodes or encrypts multi-information.." – I think it should be "metasurfaces".
2. "Minimalist metasurface" – very dubious definition.
3. "meta-transformer" – is not defined at all, but just pops up in the abstract. For me is sounds unclear, whether it is based on metamaterials/metasurfaces or it possesses meta (hyper) properties in transformation.
4. A light beam has polarization, so any scalar operations with fields' characteristics like the one in (1) should be commented. Also, what about the phase – how it is taken into account here?
5. "Under the illumination of RCP incidence..." – definitely wrong wording, incidence cannot illuminate and it has no polarization, it is light which has RCP polarization.

6. I'm wondering if in the paper describing fully classical properties of light it is important to use somewhere Planck's constant adjusted to the photon spin? Isn't the conversion of spin from $+h$ to $-h$ simply the RCP-LCP conversion?

Response to the Reviewers' Comments on the Manuscript NCOMMS-19-13458 entitled "Dielectric Multi-momentum Transformer in the Visible" submitted to Nature Communications

We would like to thank all reviewers for the careful and detailed reviewing of this work. The valuable comments have allowed us to significantly improve the manuscript. The detailed point-by-point response to the reviewers' comments is attached below. All changes have been highlighted in purple color in the revised manuscript.

Response to Reviewer #1

Comments	Optical metasurface devices, ultrathin inhomogeneous media with planar nanostructures that can manipulate light propagation in a desirable manner, leading to the development of many optical devices with unusual functionalities. Jin and colleagues propose and experimentally demonstrated a multi-momentum metasurface transformer based on the combination of intrinsic multi-phase of light, orbital angular momentum (OAM) and linear momentum. Although much efforts have been put in the generation, detection and manipulation of OAM beams, this work is interesting and can stimulate new thinking. The results presented in the manuscript seem to be correct, the experimental evidence provided support the conclusions, the manuscript is mostly clear. Regarding the significance of the results, I believe this manuscript will be of interest to specialists within the relevant fields.
Response	Thanks a lot for the positive comments!
Comments	Thus, it seems to me that the manuscript can be accepted in this journal if the comments below can be fully addressed. 1. In Fig.2b, the calculated efficiency of the metasurface is very high. However, the signal-to-noise ratio in the Fig. 3b is very bad. I am not sure about the experimental value.
Response 1.1	The high conversion efficiency in Fig. 2b is from experimental results. The lower signal-to-noise ratio is caused by smaller phase difference between the incident vortex beams and encoded more OAM states. The details are

	discussed in the Response 1.2.
Comments	2. The performance of the devices. In Fig. 2d, the experimental results are better than those in Fig.3b. Is this related to the selection of the OAM? For example, the two topological charges with same absolute values but opposite signs ($l=-5$ and $+5$) are better than two positive values (1 and 3) or negative values.
Response 1.2	The qualities of experimental results are related to the difference of the encoded OAM topological charges but not limited to be conjugate state (same absolute values but opposite signs). The performance of OAM meta-transformer can be improved with the increase of the phase difference among the incident vortex beams $\Delta\psi_{OAM}$. The topological charges difference requires the phase difference $10\theta(x_0, y_0)$ in Fig. 2d but $3\theta(x_0, y_0)$ in Fig. 3b. Therefore, the experimental results in Fig. 2d has better performance than those in Fig. 3b.
Comments	3. The crosstalk issue. In Fig.3b, I can see other patterns in the OAM selective patterns. How many patterns can be encoded in the metasurface? I am sure there is a limit.
Response 1.3	The capability of a single meta-transformer is affected by many conditions, such as the number of unit cell and momentum-state region. To have a more specific and valid discussion on this issue, we restrict some conditions: the number of unit cell is fixed at 600×600; OAM states $l^{(n)}$ range from -8 to 8; and LM states are in visible region. Under the given conditions, a single meta-transformer is able to support 5 OAM states or 6 LM states. By introducing the polarization-dependent response of TiO_2 nano-fins, the capability of a meta-transformer can increase to 9 OAM states or 11 LM states. In the revised manuscript, we have discussed these results in detail on the Page 11 as follows: In manuscript: The capability is a critical issue for the multi-tasked metasurface. Based on the multi-momentum phase retrieval algorithm provided in this work, a single meta-transformer is able to support 5 OAM states ranging from -8 to 8 or 6 LM states in visible region (Supplementary note 8). With the help of the polarization-dependent response of TiO_2 nano-fins, the number of patterns encoded in a meta-transformer can increase to 9 for OAM or 11 for LM. Furthermore, the capability of a meta-transformer can also be improved by increasing the number of units, and widening the multi-momentum state region. Besides, by considering the donut shapes of vortex beams, introducing random phase mask, and combining the LM and OAM simultaneously in the phase retrieval algorithm, the multi-momentum meta-transformer has the potential to remarkably increase the carried information states.

In supplementary information:

8. The capability of multi-momentum meta-transformer

In this simulation, the number of units in a phase profile is fixed at 600×600 .

Supplementary Figure 5| The capability of an OAM meta-transformer.

(a) The OAM meta-transformer designed for the beam with RCP. This meta-transformer encodes 5 states and $l^{(n)}$ are from -8 to 8 with step 4. (b) The OAM meta-transformer designed for the beam with CP. With the help of polarization, a single OAM meta-transformer is able to reconstruct 9 patterns with $l^{(n)}$ ranging from -8 to 8.

Supplementary Figure 6| The capability of a LM meta-transformer.

(a) The LM meta-transformer designed for the beam with RCP. In the visible region, a single LM meta-transformer is capable of supporting 6 LM states. (b) The LM meta-transformer designed for the beam with CP. The capability of a single LM meta-transformer can be increased to 11 LM states.

Comments	4. The generation of OAM beams is very important for device characterization. More detailed specifications of the SLM should be provided.
Response 1.4	In this work, the OAM beams are generated by fork gratings. The fork grating is fork-shaped binary computer-generated hologram, which produces a diffracted beam with helical phasefront. To specify the generation of OAM beam, we have added the description of fork grating and a reference in Supplementary Note 6 as follow: The fork grating, fork-shaped binary computer-generated hologram³, is employed here to generate vortex beams. Reference: 3. Stoyanov, L., Topuzoski, S., Stefanov, I., Janicijevic, L. & Dreischuh, A. Far field diffraction of an optical vortex beam by a fork-shaped grating. Optics Communications 350, 301-308 (2015).
Comments	5. The reconstructed color images in Figure 4 and Supplementary Figure 4 are not attractive. I would like to encourage the authors to comment on their results. Our readers deserve to know them.
Response 1.5	We have rewritten the discussion part on the reconstructed color images and discussed these results on the Page 10 as follows: Based on k_0-dependent impulse response, the LM meta-transformer generates LM-dependent patterns on the identical position (Fig. 2(e) and (f)). Under illumination of red, green and blue beams simultaneously, these generated patterns overlapping each other enable the realization of colorful holographic images (Supplementary note 7). The colorful holographic images contain not only three-primary colors (RGB) but also their secondary colors (cyan, magenta, yellow, as well as white). Moreover, the LM meta-transformer can also support complex patterns with gradient color (in Fig. 4 (a) and (b)). Figure 4 (b) reports the simulation and experimental results for each color and their superposition. These results indicate the accurate spatial control of the reconstructed images, and the crosstalk among different LMs is eliminated. The key feature for such gradient color image is that the color gamut spans the whole color triangle. Therefore, the LM meta-transform has the capability to display vivid colorful image.
Comments	6. The overall language in the manuscript should be polished. There are also some typos, which need to be corrected.
Response 1.6	We have corrected the typos and polished the language in the manuscript.

Response to Reviewer #2

Comments	All-dielectric metasurfaces are in a big trend now towards ultimate control over the polarization state, focus points, orbital angular momentum, etc. of light beams. What is important that the thickness of the metasurfaces is comparative or even less than the wavelength, providing ultracompact tools for light manipulation. The paper well advances the subject of light control reporting on multiplication of actions: now orbital angular momentum can be combined with light's linear momentum. Certainly, such idea confirmed in direct experiments warrants publication in Nature Communications.
Response	Thanks a lot for the positive comments!
Comments	I found few not clear wording and explanations/definitions that should be improved for the sake of the readers. Otherwise I don't need to review it again after such minor corrections. 1. "The majority of multi-tasked metasurface encodes or encrypts multi-information.." – I think it should be "metasurfaces"..
Response 2.1	We have replaced "multi-tasked metasurface" with "multi-tasked metasurfaces" as follow: The majority of multi-tasked metasurfaces encode or encrypt multi-information either into the carefully tailored metasurfaces or in pre-set complex incident beam arrays.
Comments	2. "Minimalist metasurface" – very dubious definition.
Response 2.2	We have deleted this definition and rewritten related sentences as follow: Here, we propose and demonstrate a multi-momentum transformation metasurface (i.e., meta-transformer), by fully synergizing intrinsic properties of light, e.g., orbital angular momentum (OAM) and linear momentum (LM), with a fixed phase profile imparted by a metasurface .
Comments	3. "meta-transformer" – is not defined at all, but just pops up in the abstract. For me is sounds unclear, whether it is based on metamaterials/metasurfaces or it possesses meta (hyper) properties in transformation.
Response 2.3	The "meta-transformer" means that the transformer is based on metasurface. To avoid ambiguity, we have added the definition in abstract: "Here, we propose and demonstrate a multi-momentum transformation metasurface (i.e., meta-transformer),"

Comments	4. A light beam has polarization, so any scalar operations with fields' characteristics like the one in (1) should be commented. Also, what about the phase – how it is taken into account here?
Response 2.4	We have added the discussion on polarization and phase in the expression of diffraction field. When polarization and phase are taken into account, the diffraction field can be expressed as $\mathbf{U}(x, y, z) = \iint_{-\infty}^{+\infty} \mathbf{U}_{meta}^{T/R}(x_0, y_0) \mathbf{U}_{inc}(x_0, y_0) h(x - x_0, y - y_0, z) dx_0 dy_0,$ $\mathbf{U}_{meta}^{T/R}(x_0, y_0) \mathbf{U}_{inc}(x_0, y_0) = \begin{bmatrix} u_{x_0 x_0}^{T/R}(x_0, y_0) & u_{x_0 y_0}^{T/R}(x_0, y_0) \\ u_{y_0 x_0}^{T/R}(x_0, y_0) & u_{y_0 y_0}^{T/R}(x_0, y_0) \end{bmatrix} \begin{bmatrix} E_{x_0}(x_0, y_0) \\ E_{y_0}(x_0, y_0) \end{bmatrix}.$ where $u_{mn}^{T/R}$ (m, n denotes x_0, y_0) is complex number. In consideration of readability, we put this discussion in Supplementary note 1 and 2.
Comments	5. “Under the illumination of RCP incidence...” – definitely wrong wording, incidence cannot illuminate and it has no polarization, it is light which has RCP polarization.
Response 2.5	We have rewritten this sentence as “Under the illumination of right circularly polarized beam.....”.
Comments	6. I’m wondering if in the paper describing fully classical properties of light it is important to use somewhere Plank’s constant adjusted to the photon spin? Isn’t the conversion of spin from +h to –h simply the RCP-LCP conversion?
Response 2.6	We agree with the referee that the conversion of spin from $+\hbar$ to $-\hbar$ is different from the RCP-LCP. Thus, we have replaced the “ $\pm\hbar$ ” with circular polarizations.

Reviewers' Comments:

Reviewer #1:

Remarks to the Author:

The authors have addressed my concerns. I am happy with this version now.